# Hypertensive disorders of pregnancy and the risk of chronic kidney disease: A Swedish registry-based cohort study

Peter M. Barrett[1,2]*, Fergus P. McCarthy[2,3], Marie Evans[4], Marius Kublickas[5], Ivan J. Perry[1], Peter Stenvinkel[4], Ali S. Khashan[1,2‡], Karolina Kublickiene[4‡]

1 School of Public Health, University College Cork, Cork, Ireland, 2 Irish Centre for Maternal and Child Health Research, University College Cork, Cork, Ireland, 3 Department of Obstetrics & Gynaecology, Cork University Maternity Hospital, Cork, Ireland, 4 Division of Renal Medicine, Department of Clinical Intervention, Science and Technology (CLINTEC), Karolinska Institutet, Stockholm, Sweden, 5 Department of Obstetrics & Gynaecology, Karolinska University Hospital, Stockholm, Sweden

‡ These authors are joint senior authors on this work and contributed equally.
* peter.barrett@ucc.ie

## Abstract

### Background

Hypertensive disorders of pregnancy (HDP) (preeclampsia, gestational hypertension) are associated with an increased risk of end-stage kidney disease (ESKD). Evidence for associations between HDP and chronic kidney disease (CKD) is more limited and inconsistent. The underlying causes of CKD are wide-ranging, and HDP may have differential associations with various aetiologies of CKD. We aimed to measure associations between HDP and maternal CKD in women who have had at least one live birth and to identify whether the risk differs by CKD aetiology.

### Methods and findings

Using data from the Swedish Medical Birth Register (MBR), singleton live births from 1973 to 2012 were identified and linked to data from the Swedish Renal Register (SRR) and National Patient Register (NPR; up to 2013). Preeclampsia was the main exposure of interest and was treated as a time-dependent variable. Gestational hypertension was also investigated as a secondary exposure. The primary outcome was maternal CKD, and this was classified into 5 subtypes: hypertensive, diabetic, glomerular/proteinuric, tubulointerstitial, and other/nonspecific CKD. Cox proportional hazard regression models were used, adjusting for maternal age, country of origin, education level, antenatal BMI, smoking during pregnancy, gestational diabetes, and parity. Women with pre-pregnancy comorbidities were excluded.

The final sample consisted of 1,924,409 women who had 3,726,554 singleton live births. The mean (±SD) age of women at first delivery was 27.0 (±5.1) years. Median follow-up was 20.7 (interquartile range [IQR] 9.9–30.0) years. A total of 90,917 women (4.7%) were diagnosed with preeclampsia, 43,964 (2.3%) had gestational hypertension, and 18,477 (0.9%) developed CKD. Preeclampsia was associated with a higher risk of developing CKD during

**Data Availability Statement:** Data are from the Swedish Medical Birth Register, National Patient Register and Swedish Renal Register. Data cannot be put into a public data repository due to Swedish

confidentiality regulations for registry data. Details on the application procedures for data usage is available on the home pages of the respective registries: the Medical Birth Register (https://www.socialstyrelsen.se/en/statistics-and-data/registers/alla-register/the-swedish-medical-birth-register/); the National Patient Register (https://www.socialstyrelsen.se/en/statistics-and-data/registers/alla-register/the-national-patient-register/); and the Swedish Renal Register (https://www.medscinet.net/snr/). Information on how to access the data can also be found using these contact details: Phone +46(0)75-247 30 00, email socialstyrelsen@socialstyrelsen.se.

**Funding:** This work was performed within the Irish Clinical Academic Training (ICAT) Programme, supported by the Wellcome Trust and the Health Research Board (Grant Number 203930/B/16/Z), the Health Service Executive National Doctors Training and Planning, and the Health and Social Care, Research and Development Division, Northern Ireland. PMB is employed as an ICAT Fellow. KK is supported by the Strategic Research Programme in Diabetes at Karolinska Institutet (Swedish Research Council grant number 2009-1068 and Swedish Research Council grant number 2018-00932), Stockholm County Council (ALF), the Swedish Kidney Foundation (Njurfonden). The funders had no role in study design, data collection and analysis, decision to publish, or preparation of the manuscript.

**Competing interests:** I have read the journal's policy and the authors of this manuscript have the following competing interests: ME has participated in advisory board meetings (Astellas, Astra Zeneca, Vifor Pharma) and has received payment for lectures (Astellas, Vifor Pharma).

**Abbreviations:** CKD, chronic kidney disease; CVD, cardiovascular disease; ESKD, end-stage kidney disease; HDP, hypertensive disorders of pregnancy; HELLP, Haemolysis, Elevated Liver enzymes, Low Platelets; ICD, International Classification of Diseases; IQR, interquartile range; LMP, last menstrual period; MBR, Medical Birth Register; NPR, National Patient Register; NRF-2, nuclear factor erythroid 2–related factor 2; PPV, positive predictive value; SGA, small for gestational age; SLE, systemic lupus erythematosus; SRR, Swedish Renal Register; STROBE, the Strengthening the Reporting of Observational Studies in Epidemiology.

follow-up (adjusted hazard ratio [aHR] 1.92, 95% CI 1.83–2.03, $p < 0.001$). This risk differed by CKD subtype and was higher for hypertensive CKD (aHR 3.72, 95% CI 3.05–4.53, $p < 0.001$), diabetic CKD (aHR 3.94, 95% CI 3.38–4.60, $p < 0.001$), and glomerular/proteinuric CKD (aHR 2.06, 95% CI 1.88–2.26, $p < 0.001$). More modest associations were observed between preeclampsia and tubulointerstitial CKD (aHR 1.44, 95% CI 1.24–1.68, $p < 0.001$) or other/nonspecific CKD (aHR 1.51, 95% CI 1.38–1.65, $p < 0.001$). The risk of CKD was increased after preterm preeclampsia, recurrent preeclampsia, or preeclampsia complicated by pre-pregnancy obesity. Women who had gestational hypertension also had increased risk of developing CKD (aHR 1.49, 95% CI 1.38–1.61, $p < 0.001$). This association was strongest for hypertensive CKD (aHR 3.13, 95% CI 2.47–3.97, $p < 0.001$). Limitations of the study are the possibility that cases of CKD were underdiagnosed in the national registers, and some women may have been too young to have developed symptomatic CKD despite the long follow-up time. Underreporting of postpartum hypertension is also possible.

## Conclusions

In this study, we found that HDP are associated with increased risk of maternal CKD, particularly hypertensive or diabetic forms of CKD. The risk is higher after preterm preeclampsia, recurrent preeclampsia, or preeclampsia complicated by pre-pregnancy obesity. Women who experience HDP may benefit from future systematic renal monitoring.

## Author summary

### Why was this study done?

- Preeclampsia is associated with increased long-term risk of heart disease and end-stage kidney disease (ESKD; requiring dialysis or transplant) in women.

- Less is known about the long-term risk of chronic kidney disease (CKD) following preeclampsia, although it is much more common than ESKD.

- There are many different types of CKD, and we sought to identify whether preeclampsia was equally associated with different subtypes of CKD. Large-scale, high-quality datasets with long periods of follow-up are required to investigate this.

### What did the researchers do and find?

- We used nationally representative data from 1.9 million women (3.7 million live births) in Sweden to measure the risk of CKD following preeclampsia over a 41-year period.

- We controlled our analysis for multiple confounding factors, including maternal age, country of origin, education level, BMI, smoking, and gestational diabetes. We excluded women with underlying medical conditions who were at higher risk of CKD at baseline.

- Women who had preeclampsia had almost double the risk of developing any CKD during follow-up compared to women with no preeclampsia. They also had more than 3

times higher risk of CKD linked to high blood pressure (hypertensive CKD) or diabetes specifically.

- Women who experienced preterm preeclampsia, recurrent preeclampsia, or preeclampsia on a background of pre-pregnancy obesity were at highest risk of CKD.

### What do these findings mean?

- Our findings suggest that women with a history of preeclampsia are at increased risk of long-term CKD, particularly hypertensive or diabetic forms of CKD.

- Women who experience preeclampsia may benefit from future systematic renal monitoring.

## Introduction

Preeclampsia is characterised by the development of de novo hypertension after 20 weeks' gestation, in the presence of either proteinuria, maternal organ dysfunction (including renal insufficiency), or evidence of foetal growth restriction [1]. It complicates 3%–5% of pregnancies worldwide [2], and affected women are at higher risk of long-term cardiovascular disease (CVD) [3–5]. Preeclampsia has also been described as a reversible kidney disease that typically self-resolves within 3 months of delivery [6]. However, there is increasing evidence that some women experience sustained renal dysfunction, and large cohort studies have reported an increased risk of end-stage kidney disease (ESKD) [7–10]. Biological mechanisms are uncertain; this may be due to lasting vascular endothelial dysfunction related to elevated levels of anti-angiogenic proteins—such as soluble fms-like tyrosine kinase-1—or it may be due to direct glomerular damage related to underexpression of nuclear factor erythroid 2–related factor 2 (NRF-2) [11–13]. Other changes in the renin-angiotensin-aldosterone system, metabolic system, and factors causing endothelial dysfunction may also be involved.

Chronic kidney disease (CKD) is much more prevalent than ESKD, and although it may be considered a precursor to ESKD, the evidence for associations between preeclampsia and CKD has been inconsistent to date [6, 14]. Cohort studies from Scotland and Denmark have reported increased risk of CKD following preeclampsia [15, 16], but these findings have not been replicated elsewhere [17, 18]. The underlying causes of CKD are wide-ranging, and it is plausible that hypertensive disorders of pregnancy (HDP) differentially affect the risk of CKD subtypes, but few studies have considered this when investigating associations. Moreover, while the risk of ESKD is higher among women who experience preeclampsia concurrently with preterm delivery or small for gestational age (SGA) [7, 8], it is unclear whether this is also the case for CKD.

Gestational hypertension is another common hypertensive disorder that arises de novo after 20 weeks' gestation in the absence of proteinuria and is not typically accompanied by organ dysfunction or foetal growth restriction [1]. Although gestational hypertension is regarded as an independent risk factor for subsequent CVD [19], few studies have investigated associations with CKD [14]. The aim of this study is to investigate whether HDP (preeclampsia, gestational hypertension) are associated with the long-term risk of maternal CKD, and to

identify whether the risk differs according to CKD aetiology or by concurrent preterm delivery or SGA.

## Methods

### Study population

Women who had singleton live births between January 1, 1973, and December 31, 2012, were identified from the Swedish Medical Birth Register (MBR; established 1973). The MBR contains detailed information on over 96% of births in Sweden [20]. We used hospitalisation data from the Swedish National Patient Register (NPR; established 1964) and the Swedish Renal Register (SRR; established 1991) to identify women who developed CKD during follow-up, until December 31, 2013 (study end date). Data from all registers were linked using the anonymised unique national identification number, which is issued to all citizens of Sweden. Data from the Swedish Death Register and Migration Register were also available until December 31, 2013, and were used for censoring. We excluded multiple pregnancies (*n* = 148,339) and pregnancies with implausible dates of delivery (*n* = 312) from all analyses at baseline. We also excluded women who had stillbirths (*n* = 14,107) to avoid potential confounding since they are more likely to experience preeclampsia compared with women who only have live births, and they are at increased risk of long-term CKD [21] (S1 Fig).

We identified women with pre-pregnancy CKD, ESKD, CVD, chronic hypertension, diabetes (type 1 or 2), systemic lupus erythematosus (SLE), systemic sclerosis, coagulopathies, hemoglobinopathies, or vasculitides from the MBR, and they were excluded at baseline. We also excluded all women in the NPR who were admitted to hospital with any of those diagnoses before their first date of delivery. We used 3 iterations of International Classification of Diseases (ICD) coding to identify pre-existing diseases in the NPR: ICD-8 coding from 1973 to 1986, ICD-9 coding from 1987 to 1996, and ICD-10 coding from 1997 to 2013. The full list of ICD codes used in the study is summarised in S1 Table. Furthermore, we identified hospital admissions and outpatient reviews for preeclampsia and gestational diabetes in the NPR and used this information to supplement data in the MBR.

This study is reported as per the Strengthening the Reporting of Observational Studies in Epidemiology (STROBE) guideline (S2 Table). All data were anonymised and nonidentifiable. Ethical approval was granted by the Swedish Ethical Review Authority in Stockholm (Regionala Etikprovningsnamnden Stockholm) (Dnr 2012/397-31/1) and by the Social Research and Ethics Committee, University College Cork (2019–109).

### Exposure variables

Preeclampsia was the main exposure of interest and was identified in the MBR and NPR using ICD codes. Preeclampsia was defined as a diastolic blood pressure of >90 mmHg with proteinuria (≥0.3 g/day or ≥1+ on a urine dipstick) [8]. We included cases of eclampsia and Haemolysis, Elevated Liver enzymes, Low Platelets (HELLP) syndrome with preeclampsia because these conditions are rare in Sweden and there were too few affected women to allow for separate groups. However, we excluded women who developed preeclampsia superimposed on chronic hypertension, since women with pre-pregnancy hypertension were excluded at baseline.

Diagnoses of preeclampsia in the MBR have been validated previously and have high positive predictive value (PPV) for ICD-9 coded diagnoses when compared with medical records but lower PPV for ICD-8 coded diagnoses [22]. In the current study, all analyses included women who had singleton live births with or without preeclampsia from 1973 onwards. Analyses were repeated after restricting the study population to those with a first live birth from

1987 onwards (thus restricted to ICD-9 and ICD-10 coded diagnoses alone). Since 2014, proteinuria is no longer a requirement for a preeclampsia diagnosis in Sweden [1]. However, because our study ended on December 31, 2013, this did not affect our analysis.

Preeclampsia was included in statistical models as a time-dependent variable. Women were considered unexposed (i) if they never developed preeclampsia, or (ii) from the date of their index delivery (without preeclampsia) until the date of delivery of their first preeclamptic pregnancy. Women were considered 'exposed' from the date of their first preeclamptic delivery onwards, irrespective of subsequent pregnancy outcomes. For example, if a woman had 3 live births and only experienced preeclampsia in her second pregnancy, she was considered unexposed between delivery 1 and delivery 2 but was considered exposed from delivery 2 onwards (despite the non-preeclamptic third pregnancy).

In accordance with international guidelines, we did not classify preeclampsia as mild or severe disease [1]. However, preeclampsia was considered together with SGA and preterm delivery respectively, and these may be considered proxy markers of severity [23]. For SGA, a series of dummy variables were included to represent nonoverlapping scenarios: (i) preeclampsia alone, (ii) SGA alone, or (iii) preeclampsia and SGA (co-occurring). SGA was defined in the MBR as a birth weight of 2 SDs below the sex-specific and gestational age distributions, according to Swedish weight-based growth standards [24], and was treated as a time-dependent variable.

Preterm delivery was defined as any delivery before 37 weeks' gestation. This was largely estimated based on second trimester ultrasound (from 1982 onwards) but was estimated from maternal report of last menstrual period (LMP) prior to that [20]. Preterm deliveries were categorised as moderate (32 weeks to 36+6 weeks), very (28 weeks to 31+6 weeks), or extremely preterm (<28 weeks gestation). The latter two categories (very/extremely preterm) were combined in analyses due to small numbers. Maternal exposure to preterm delivery was time dependent and was allowed to change multiple times across different pregnancies, but exposure status was always based on the earliest gestation of any previous delivery.

Furthermore, we considered the effect of recurrent preeclampsia on CKD risk among women who had exactly 2 deliveries. We categorised these women as follows: (1) no preeclampsia, (2) preeclampsia in one pregnancy, and (3) preeclampsia in both pregnancies.

Gestational hypertension was a secondary exposure variable of interest and was defined as blood pressure of at least 140/90 mm Hg (in at least 2 readings 6 or more hours apart), without proteinuria, occurring after 20 weeks' gestation up to the date of delivery. It was included in statistical models as a time-dependent variable.

## Outcome variables

**CKD.** Maternal CKD was the primary outcome. This was defined by a recorded diagnosis of CKD in the SRR or based on a primary or secondary diagnosis of CKD in the NPR (using ICD codes). The earliest date at which a woman appeared in either the SRR or NPR was taken as her date of diagnosis, and she was censored at that date irrespective of subsequent deliveries. Women who had an identifiable congenital or genetic cause of CKD were excluded at baseline. We only considered women who were diagnosed with CKD at least 3 months after the last pregnancy, to avoid any potential misclassification with acute kidney injury or any transient renal dysfunction related to preeclampsia.

We categorised CKD diagnosis in broad aetiologies; categories were selected a priori based on guidance from the National Kidney Foundation [25], prior research [8, 16], and clinical advice from consultant nephrologists. The following categories were used: tubulointerstitial CKD, glomerular/proteinuric CKD, hypertensive CKD, diabetic CKD, and other/unspecified

CKD. The ICD codes used to define CKD in each category are shown in S1 Table. CKD sub-type/aetiology was always based on the initial CKD diagnosis, when each woman first appeared in either the SRR or NPR.

**Covariates.** The following covariates were selected a priori and adjusted for: maternal age, country of origin, education level, antenatal BMI at first pregnancy, smoking during pregnancy, gestational diabetes, parity, and gestational hypertension. Information on the mother's highest level of educational achievement was available from the Swedish Education Register. Maternal smoking was based on any reported smoking during pregnancy, either at first antenatal visit or at 30–32 weeks' gestation. Maternal BMI was measured at first antenatal visit. Smoking status and BMI only became available from 1982 onwards and were more complete after 1987. Missing indicator variables were created to control for missing data on smoking and BMI.

Maternal exposure to gestational diabetes and gestational hypertension were time-dependent covariates, where women were considered exposed from their date of first delivery with gestational diabetes or gestational hypertension respectively. In the analysis of gestational hypertension and maternal CKD, we adjusted for preeclampsia as a time-dependent covariate.

## Statistical analysis

Each woman's entry date in the study was the date of her first live birth. The association between preeclampsia and risk of maternal CKD was estimated using the Kaplan-Meier method. We used multivariable Cox proportional hazard regression models to estimate age-adjusted and fully adjusted hazard ratios (aHRs) and 95% CIs for the associations between preeclampsia and maternal CKD. We followed women from date of entry until date of CKD diagnosis or study end date (December 31, 2013), whichever came first. Thus, women stopped contributing person-time once they were diagnosed with CKD, and any subsequent pregnancies were not included in the analysis. Women who died or emigrated during follow-up were censored on that date. Thus, the reported HRs are the HRs that would be seen if mortality could be eliminated during the study period [26]. We used log cumulative hazard plots to ensure that the proportional hazards assumption was met.

Our analysis of preeclampsia and CKD was pre-planned using 4 separate models. Model 1 explored the association between any preeclampsia and CKD (versus women who never had preeclampsia). Model 2 explored the association between preeclampsia ± SGA and CKD. Model 3 explored the association between preeclampsia ± preterm delivery and CKD. Model 4 explored the association between recurrent preeclampsia and CKD among women who had 2 live births during the study period. In each model, we considered all diagnoses of CKD collectively (overall CKD) and considered the 5 subtypes of CKD separately. Models were first adjusted for age, and then adjusted fully for all relevant covariates. In the analysis of each CKD subtype, follow-up stopped when the woman received her first diagnosis of CKD.

We conducted 3 pre-planned sensitivity analyses. Firstly, we restricted the dataset to women whose first birth occurred from 1987 onwards, when NPR coverage was more complete. The PPV for preeclampsia and gestational hypertension diagnoses was higher from this time, and information on maternal BMI and smoking was more comprehensive [20, 22]. Secondly, we categorised all births with information on maternal BMI (from 1982 onwards) according to whether mothers were obese or nonobese at the time of delivery. Thirdly, we explored the effect of excluding women who developed postpartum hypertension, to establish whether associations between preeclampsia and CKD persisted among women who remained normotensive after their last pregnancy. We undertook this analysis for all subtypes except for hypertensive CKD.

Finally, we investigated associations between gestational hypertension and CKD (versus women who never had gestational hypertension). Again, we considered all diagnoses of CKD collectively (overall CKD) and separate CKD subtypes. All analyses were performed using Stata version 15 (StataCorp, College Station, Texas).

## Results

The study cohort consisted of 1,924,409 unique women who had 3,726,554 singleton live births, followed up for a total of 42,118,889 person-years. The mean age at first delivery was 27.0 (± SD 5.1) years, and median follow-up time was 20.7 years (interquartile range [IQR] 9.9–30.0 years). There were 53,265 deaths (2.8%).

There were 90,917 women (4.7%) diagnosed with preeclampsia at least once (Table 1). They were more likely to be native Swedes, overweight or obese, more likely to have experienced other adverse pregnancy outcomes (preterm delivery, SGA, or gestational diabetes), and less likely to be smokers compared to women who never experienced preeclampsia.

From 1973 to 2013, 18,477 women (0.9%) developed CKD, of whom 2,813 (15.2%) had tubulointerstitial CKD, 6,068 (32.8%) had glomerular/proteinuric CKD, 797 (4.3%) had hypertensive CKD, 1,226 (6.6%) had diabetic CKD, and 7,573 (41.0%) had CKD due to other/ unspecified causes. The median time to CKD diagnosis after first live birth (overall) was 16.8 years (IQR 7.2–26.5). The median time to CKD diagnosis varied by aetiology: tubulointerstitial CKD: 14.6 years (IQR 5.8–24.2); glomerular/proteinuric CKD: 10.9 years (IQR 4.4–18.5); hypertensive CKD: 22.2 years (IQR 15.0–30.1); diabetic CKD: 22.1 years (IQR 12.5–29.4); and other/unspecified CKD: 22.1 years (IQR 11.5–30.0). For all CKD aetiologies, the median time to diagnosis was significantly shorter in women who had previous preeclampsia and was shortest for glomerular/proteinuric CKD (median 7.7 years; IQR 2.0–15.7) (S3 Table).

### Preeclampsia

Women who had ever experienced preeclampsia were at higher risk of developing CKD compared with women who never had preeclampsia (aHR 1.92, 95% CI 1.83–2.03, $p < 0.001$). This risk differed by CKD subtype and was highest for hypertensive CKD (aHR 3.72, 95% CI 3.05–4.53, $p < 0.001$), diabetic CKD (aHR 3.94, 95% CI 3.38–4.60, $p < 0.001$), and glomerular/proteinuric CKD (aHR 2.06, 95% CI 1.88–2.26, $p < 0.001$). The risk was lower for other/unspecified CKD (aHR 1.51, 95% CI 1.38–1.65, $p < 0.001$) and tubulointerstitial CKD (aHR 1.44, 95% CI 1.24–1.68, $p < 0.001$).

There was little difference in CKD risk between preeclamptic women who experienced concurrent SGA and those who did not (Table 2). Hypertensive CKD risk was the exception and was more likely in women who had preeclampsia + SGA (versus neither, aHR 5.23, 95% CI 3.51–7.79, $p < 0.001$). Women who had SGA alone (without preeclampsia) were also at higher risk of CKD, but the associations were less marked than for preeclampsia. Women who had preeclampsia and who delivered at earlier gestation also had higher risk of CKD (Table 3). Women with at least one preeclamptic delivery before 32 weeks' gestation were at particularly high risk of CKD (versus normal term deliveries, aHR 3.19, 95% CI 2.53–4.02, $p < 0.001$).

A total of 855,095 women in the sample had only 2 births, of whom 4.4% ($n = 37,322$) had preeclampsia once and 0.5% ($n = 4,335$) had preeclampsia twice (recurrent preeclampsia). Compared with women who never had preeclampsia, those who had recurrent preeclampsia had the greatest risk of developing any form of CKD (aHR 2.64, 95% CI 2.14–3.25, $p < 0.001$) (Table 4). Again, the risk was stronger for hypertensive CKD (aHR 5.30, 95% CI 2.47–11.36, $p < 0.001$), diabetic CKD (aHR 6.80, 95% CI 3.96–11.68, $p < 0.001$), and glomerular/proteinuric CKD (aHR 3.42, 95% CI 2.44–4.78, $p < 0.001$).

**Table 1. Maternal characteristics and pregnancy outcomes among women who had live births between 1973 and 2012 in Sweden, stratified by exposure to pre-eclampsia (N = 1,924,409).**

| | No preeclampsia, *n* (%) *n* = 1,833,492 (95.3%) | Preeclampsia, *n* (%) *n* = 90,917 (4.7%) | *p*-Value |
|---|---|---|---|
| **Age at first pregnancy (years)** | | | *p* < 0.001 |
| <20 | 103,561 (5.7) | 5,971 (6.6) | |
| 20–29 | 1,184,635 (64.6) | 59,030 (64.9) | |
| 30–39 | 520,110 (28.4) | 24,424 (26.9) | |
| ≥40 | 25,186 (1.4) | 1,492 (1.6) | |
| **Native country** | | | *p* < 0.001 |
| Sweden | 1,550,745 (84.6) | 80,663 (88.7) | |
| Elsewhere | 282,747 (15.4) | 10,254 (11.3) | |
| **Education level** | | | *p* < 0.001 |
| Less than upper secondary | 243,022 (13.3) | 11,571 (12.7) | |
| Upper secondary | 825,165 (45.0) | 44,073 (48.5) | |
| Third level | 726,169 (39.6) | 34,078 (37.5) | |
| Missing | 39,136 (2.1) | 1,195 (1.3) | |
| **BMI in early pregnancy (kg/m$^2$)** | | | *p* < 0.001 |
| Underweight: <18.5 | 44,069 (2.4) | 1,391 (1.5) | |
| Normal: 18.5–24.9 | 676,382 (36.9) | 28,187 (31.0) | |
| Overweight: 25–29.9 | 179,529 (9.8) | 12,669 (13.9) | |
| Obesity: ≥30 | 62,487 (3.4) | 7,225 (8.0) | |
| Missing | 871,025 (47.5) | 41,445 (45.6) | |
| **Maternal smoking** | | | *p* < 0.001 |
| No | 971,826 (53.0) | 52,761 (58.0) | |
| Yes | 198,463 (10.8) | 7,704 (8.5) | |
| Missing | 663,203 (36.2) | 30,452 (33.5) | |
| **Gestational diabetes (ever)** | | | *p* < 0.001 |
| No | 1,817,243 (99.1) | 88,935 (97.8) | |
| Yes | 16,249 (0.9) | 1,982 (2.2) | |
| **Preterm delivery (ever)** | | | *p* < 0.001 |
| No | 1,695,439 (92.5) | 70,383 (77.4) | |
| Yes | 138,188 (7.5) | 20,581 (22.6) | |
| **SGA (ever)** | | | *p* < 0.001 |
| No | 1,747,083 (95.4) | 77,094 (84.9) | |
| Yes | 84,540 (4.6) | 13,751 (15.1) | |
| **Decade of first birth** | | | *p* < 0.001 |
| 1973–1979 | 490,813 (26.8) | 21,231 (23.4) | |
| 1980–1989 | 400,822 (21.9) | 21,524 (23.7) | |
| 1990–1999 | 399,164 (21.8) | 21,950 (24.1) | |
| 2000–2012 | 542,693 (29.6) | 26,212 (28.8) | |

**Abbreviation:** SGA, small for gestational age

## Sensitivity analysis

When the dataset was restricted to first deliveries after 1987, some associations were strengthened. However, the overall results were not substantially different (S4–S6 Tables). When births were stratified by maternal obesity at index pregnancy (from 1982), the association between preeclampsia and CKD was stronger in obese women (aHR 2.27, 95% CI 1.92–2.69) than it

**Table 2. HRs for maternal CKD by history of preeclampsia and SGA, among women who had live births between 1973 and 2012 in Sweden (N = 1,924,409).**

| | | CKD, n | Age-adjusted HR (95% CI*) | Fully adjusted HR (95% CI*) |
|---|---|---|---|---|
| **Overall CKD** | | | | |
| No preeclampsia, no SGA | | 15,783 | 1.0 | 1.0 |
| Preeclampsia only | | 1,318 | 2.08 (1.96–2.20) | 1.96 (1.85–2.08) |
| SGA only | | 1,150 | 1.45 (1.36–1.53) | 1.32 (1.24–1.40) |
| Preeclampsia and SGA | | 226 | 2.11 (1.85–2.41) | 1.95 (1.71–2.22) |
| **1.** | **Tubulointerstitial CKD** | | | |
| | No preeclampsia, no SGA | 2,458 | 1.0 | 1.0 |
| | Preeclampsia only | 157 | 1.52 (1.29–1.78) | 1.47 (1.25–1.73) |
| | SGA only | 171 | 1.45 (1.24–1.70) | 1.30 (1.11–1.51) |
| | Preeclampsia and SGA | 27 | 1.51 (1.03–2.20) | 1.41 (0.97–2.07) |
| **2.** | **Glomerular/proteinuric CKD** | | | |
| | No preeclampsia, no SGA | 5,151 | 1.0 | 1.0 |
| | Preeclampsia only | 425 | 2.10 (1.90–2.32) | 2.11 (1.90–2.33) |
| | SGA only | 417 | 1.61 (1.45–1.77) | 1.46 (1.32–1.62) |
| | Preeclampsia and SGA | 75 | 2.22 (1.76–2.78) | 2.16 (1.71–2.71) |
| **3.** | **Hypertensive CKD** | | | |
| | No preeclampsia, no SGA | 610 | 1.0 | 1.0 |
| | Preeclampsia only | 104 | 4.43 (3.59–5.46) | 3.60 (2.90–4.47) |
| | SGA only | 57 | 1.76 (1.34–2.30) | 1.54 (1.17–2.02) |
| | Preeclampsia and SGA | 26 | 6.82 (4.60–10.11) | 5.23 (3.51–7.79) |
| **4.** | **Diabetic CKD** | | | |
| | No preeclampsia, no SGA | 954 | 1.0 | 1.0 |
| | Preeclampsia only | 189 | 5.14 (4.40–6.01) | 4.03 (3.42–4.74) |
| | SGA only | 57 | 1.16 (0.89–1.51) | 1.05 (0.80–1.37) |
| | Preeclampsia and SGA | 26 | 4.28 (2.90–6.33) | 3.49 (2.36–5.16) |
| **5.** | **Other/unspecified CKD** | | | |
| | No preeclampsia, no SGA | 6,611 | 1.0 | 1.0 |
| | Preeclampsia only | 443 | 1.65 (1.50–1.81) | 1.54 (1.40–1.70) |
| | SGA only | 448 | 1.34 (1.22–1.47) | 1.24 (1.13–1.37) |
| | Preeclampsia and SGA | 72 | 1.59 (1.26–2.00) | 1.46 (1.15–1.84) |

*All $p < 0.001$.

HRs represent separate Cox regression models for associations between preeclampsia and maternal CKD. Preeclampsia was a time-dependent variable. Fully adjusted models controlled for maternal age, country of origin, education level, parity, maternal BMI, smoking in pregnancy, exposure to gestational diabetes, and exposure to gestational hypertension. Models were stratified by year of delivery.

**Abbreviations:** CKD, chronic kidney disease; HR, hazard ratio; SGA, small for gestational age

was for nonobese women (aHR 1.71, 95 CI 1.56–1.87, $p$ for interaction $< 0.01$) (Table 5). These differences persisted in analyses of preeclampsia ± SGA or preterm delivery, respectively, but not for recurrent preeclampsia. When women who developed postpartum hypertension were excluded from analyses, most associations between preeclampsia and CKD subtypes were attenuated, but not meaningfully different (S7–S9 Tables).

## Gestational hypertension

There were 43,964 women (2.3%) diagnosed with gestational hypertension at least once. Women who had ever experienced gestational hypertension were at increased risk of

**Table 3. HRs for maternal CKD by history of preeclampsia and preterm delivery, among women who had live births between 1973 and 2012 in Sweden (N = 1,924,409).**

| | | CKD, n | Age-adjusted | Fully adjusted |
|---|---|---|---|---|
| | | | HR (95% CI*) | HR (95% CI*) |
| **Overall CKD** | | | | |
| Term delivery, no preeclampsia | | 15,134 | 1.0 | 1.0 |
| Moderate preterm delivery, no preeclampsia | | 1,552 | 1.58 (1.50–1.66) | 1.46 (1.39–1.54) |
| Very/extremely preterm delivery, no preeclampsia | | 247 | 1.86 (1.64–2.11) | 1.63 (1.44–1.85) |
| Term delivery + preeclampsia | | 1,196 | 1.98 (1.87–2.10) | 1.87 (1.76–1.99) |
| Moderate preterm delivery + preeclampsia | | 276 | 2.65 (2.34–3.00) | 2.52 (2.23–2.85) |
| Very/extremely preterm delivery + preeclampsia | | 72 | 3.34 (2.65–4.21) | 3.19 (2.53–4.02) |
| 1. | **Tubulointerstitial CKD** | | | |
| | Term delivery, no preeclampsia | 2,365 | 1.0 | 1.0 |
| | Moderate preterm delivery, no preeclampsia | 220 | 1.38 (1.20–1.58) | 1.26 (1.09–1.44) |
| | Very/extremely preterm delivery, no preeclampsia | 44 | 2.05 (1.52–2.76) | 1.75 (1.30–2.36) |
| | Term delivery + preeclampsia | 146 | 1.50 (1.26–1.77) | 1.45 (1.22–1.72) |
| | Moderate preterm delivery + preeclampsia | 24 | 1.39 (0.93–2.08) | 1.37 (0.91–2.05) |
| | Very/extremely preterm delivery + preeclampsia | 14 | 3.30 (1.95–5.58) | 3.27 (1.93–5.54) |
| 2. | **Glomerular/proteinuric CKD** | | | |
| | Term delivery, no preeclampsia | 4,982 | 1.0 | 1.0 |
| | Moderate preterm delivery, no preeclampsia | 508 | 1.59 (1.45–1.74) | 1.48 (1.35–1.62) |
| | Very/extremely preterm delivery, no preeclampsia | 78 | 1.83 (1.46–2.29) | 1.59 (1.27–1.98) |
| | Term delivery + preeclampsia | 393 | 1.98 (1.78–2.20) | 1.98 (1.78–2.20) |
| | Moderate preterm delivery + preeclampsia | 81 | 2.64 (2.12–3.29) | 2.69 (2.16–3.35) |
| | Very/extremely preterm delivery + preeclampsia | 26 | 3.87 (2.63–5.70) | 3.88 (2.64–5.71) |
| 3. | **Hypertensive CKD** | | | |
| | Term delivery, no preeclampsia | 573 | 1.0 | 1.0 |
| | Moderate preterm delivery, no preeclampsia | 77 | 2.14 (1.68–2.71) | 1.92 (1.51–2.45) |
| | Very/extremely preterm delivery, no preeclampsia | 17 | 3.50 (2.16–5.67) | 2.98 (1.84–4.84) |
| | Term delivery + preeclampsia | 104 | 4.49 (3.63–5.57) | 3.65 (2.92–4.55) |
| | Moderate preterm delivery + preeclampsia | 22 | 7.06 (4.60–10.83) | 5.47 (3.55–8.43) |
| | Very/extremely preterm delivery + preeclampsia | ** | 6.94 (2.59–18.60) | 5.74 (2.14–15.40) |
| 4. | **Diabetic CKD** | | | |
| | Term delivery, no preeclampsia | 839 | 1.0 | 1.0 |
| | Moderate preterm delivery, no preeclampsia | 150 | 2.85 (2.40–3.40) | 2.54 (2.13–3.03) |
| | Very/extremely preterm delivery, no preeclampsia | 22 | 3.12 (2.04–4.76) | 2.55 (1.66–3.89) |
| | Term delivery + preeclampsia | 155 | 4.62 (3.87–5.51) | 3.69 (3.08–4.41) |
| | Moderate preterm delivery + preeclampsia | 56 | 11.70 (8.91–15.36) | 8.80 (6.67–11.60) |
| | Very/extremely preterm delivery + preeclampsia | ** | 4.10 (1.53–10.96) | 3.18 (1.19–8.51) |
| 5. | **Other/unspecified CKD** | | | |
| | Term delivery, no preeclampsia | 6,375 | 1.0 | 1.0 |
| | Moderate preterm delivery, no preeclampsia | 597 | 1.43 (1.32–1.56) | 1.34 (1.24–1.46) |
| | Very/extremely preterm delivery, no preeclampsia | 86 | 1.52 (1.23–1.88) | 1.37 (1.10–1.69) |
| | Term delivery + preeclampsia | 398 | 1.61 (1.45–1.78) | 1.50 (1.36–1.67) |
| | Moderate preterm delivery + preeclampsia | 93 | 1.78 (1.42–2.25) | 1.68 (1.33–2.12) |

*(Continued)*

**Table 3.** (Continued)

| | | CKD, *n* | Age-adjusted | Fully adjusted |
|---|---|---|---|---|
| | | | HR (95% CI*) | HR (95% CI*) |
| | Very/extremely preterm delivery + preeclampsia | 24 | 2.64 (1.78–3.94) | 2.50 (1.68–3.74) |

*All $p < 0.001$.

**Exact number not reported as cell count ≤5.

HRs represent separate Cox regression models for associations between preeclampsia and maternal CKD. Preeclampsia was a time-dependent variable. Fully adjusted models controlled for maternal age, country of origin, education level, parity, maternal BMI, smoking in pregnancy, exposure to gestational diabetes, and exposure to gestational hypertension. Models were stratified by year of delivery.

**Abbreviations:** CKD, chronic kidney disease; HR, hazard ratio; ne, not estimable; SGA, small for gestational age

developing CKD (versus no gestational hypertension, aHR 1.49, 95% CI 1.38–1.61, $p < 0.001$) (S10 Table). The association was stronger for hypertensive CKD (aHR 3.13, 95% CI 2.47–3.97,

**Table 4. HRs for maternal CKD by history of recurrent preeclampsia, among women who had live births between 1973 and 2012 in Sweden (*N* = 855,095).**

| | | CKD, *n* | Age-adjusted | Fully adjusted |
|---|---|---|---|---|
| | | | HR (95% CI*) | HR (95% CI*) |
| **Overall CKD** | | | | |
| Two pregnancies without preeclampsia | | 6,326 | 1.0 | 1.0 |
| Two pregnancies, one episode of preeclampsia | | 551 | 1.90 (1.74–2.07) | 1.82 (1.66–1.99) |
| Two pregnancies, two episodes of preeclampsia | | 90 | 2.77 (2.25–3.41) | 2.64 (2.14–3.25) |
| 1. | **Tubulointerstitial CKD** | | | |
| | Two pregnancies without preeclampsia | 1,054 | 1.0 | 1.0 |
| | Two pregnancies, one episode of preeclampsia | 78 | 1.59 (1.26–2.00) | 1.58 (1.25–1.99) |
| | Two pregnancies, two episodes of preeclampsia | 10 | 1.77 (0.95–3.30) | 1.74 (0.93–3.25) |
| 2. | **Glomerular/proteinuric CKD** | | | |
| | Two pregnancies without preeclampsia | 2,062 | 1.0 | 1.0 |
| | Two pregnancies, one episode of preeclampsia | 183 | 1.94 (1.67–2.26) | 2.02 (1.73–2.35) |
| | Two pregnancies, two episodes of preeclampsia | 35 | 3.32 (2.38–4.64) | 3.42 (2.44–4.78) |
| 3. | **Hypertensive CKD** | | | |
| | Two pregnancies without preeclampsia | 217 | 1.0 | 1.0 |
| | Two pregnancies, one episode of preeclampsia | 38 | 3.93 (2.79–5.55) | 3.23 (2.25–4.63) |
| | Two pregnancies, two episodes of preeclampsia | 7 | 6.70 (3.16–14.23) | 5.30 (2.47–11.36) |
| 4. | **Diabetic CKD** | | | |
| | Two pregnancies without preeclampsia | 340 | 1.0 | 1.0 |
| | Two pregnancies, one episode of preeclampsia | 73 | 4.73 (3.67–6.10) | 3.74 (2.88–4.86) |
| | Two pregnancies, two episodes of preeclampsia | 14 | 8.44 (4.94–14.41) | 6.80 (3.96–11.68) |
| 5. | **Other/unspecified CKD** | | | |
| | Two pregnancies without preeclampsia | 2,651 | 1.0 | 1.0 |
| | Two pregnancies, one of episode preeclampsia | 179 | 1.47 (1.26–1.71) | 1.36 (1.16–1.58) |
| | Two pregnancies, two episodes of preeclampsia | 24 | 1.77 (1.18–2.64) | 1.64 (1.10–2.46) |

*All $p < 0.001$

HRs represent separate Cox regression models for associations between preeclampsia and maternal CKD. Preeclampsia was a time-dependent variable. Fully adjusted models controlled for maternal age, country of origin, education level, parity, maternal BMI, smoking in pregnancy, exposure to gestational diabetes, and exposure to gestational hypertension. Models were stratified by year of delivery.

**Abbreviations:** CKD, chronic kidney disease; HR, hazard ratio

**Table 5. HRs for maternal CKD by history of preeclampsia, among women whose first live birth occurred between 1982 and 2012 in Sweden, stratified by maternal obesity (n = 1,011,939).**

| | All women | Women with normal BMI | Nonobese women | Obese women | p-Value for interaction |
|---|---|---|---|---|---|
| | aHR (95% CI*) | aHR (95% CI) | aHR (95% CI) | aHR (95% CI) | |
| **Preeclampsia** | | | | | |
| No preeclampsia | 1.0 | 1.0 | 1.0 | 1.0 | p < 0.01 |
| Preeclampsia (any) | 1.83 (1.66–1.95) | 1.62 (1.43–1.82) | 1.71 (1.56–1.87) | 2.27 (1.92–2.69) | |
| **Preeclampsia and SGA** | | | | | |
| No preeclampsia, no SGA | 1.0 | 1.0 | 1.0 | 1.0 | p < 0.01 |
| Preeclampsia only | 1.85 (1.70–2.02) | 1.63 (1.43–1.86) | 1.75 (1.59–1.94) | 2.35 (1.96–2.81) | |
| SGA only | 1.24 (1.13–2.04) | 1.19 (1.05–1.35) | 1.21 (1.09–1.34) | 1.58 (1.15–2.17) | |
| Preeclampsia and SGA | 1.66 (1.35–2.04) | 1.61 (1.22–2.13) | 1.58 (1.25–1.98) | 2.20 (1.41–3.45) | |
| **Preeclampsia and preterm delivery** | | | | | |
| Term delivery, no preeclampsia | 1.0 | 1.0 | 1.0 | 1.0 | p < 0.01 |
| Moderate preterm delivery, no preeclampsia | 1.38 (1.28–1.50) | 1.38 (1.25–1.53) | 1.35 (1.24–1.47) | 1.65 (1.31–2.09) | |
| Very/extremely preterm delivery, no preeclampsia | 1.46 (1.18–1.79) | 1.34 (1.01–1.77) | 1.45 (1.16–1.81) | 1.53 (0.86–2.72) | |
| Term delivery + preeclampsia | 1.71 (1.55–1.88) | 1.52 (1.32–1.75) | 1.65 (1.49–1.84) | 2.05 (1.68–2.51) | |
| Moderate preterm delivery + preeclampsia | 2.26 (1.89–2.70) | 2.05 (1.58–2.66) | 2.08 (1.68–2.56) | 3.15 (2.22–4.46) | |
| Very/extremely preterm delivery + preeclampsia | 3.56 (2.67–4.74) | 3.74 (2.48–5.63) | 3.36 (2.40–4.71) | 4.68 (2.70–8.12) | |
| **Recurrent preeclampsia** | | | | | |
| Two pregnancies without preeclampsia | 1.0 | 1.0 | 1.0 | 1.0 | p = 0.469 |
| Two pregnancies, one episode of preeclampsia | 1.72 (1.52–1.96) | 1.46 (1.20–1.77) | 1.67 (1.44–1.93) | 2.09 (1.58–2.77) | |
| Two pregnancies, two episodes of preeclampsia | 2.13 (1.55–2.94) | 2.13 (1.30–3.49) | 2.20 (1.52–3.20) | 2.11 (1.12–3.98) | |

Results were based on pregnancies for which data on BMI at first antenatal visit were available. BMI was only collected from 1982 onwards in the MBR. Analysis of recurrent preeclampsia was restricted to women who had two singleton live births from 1982 to 2012 inclusive, n = 482,845.

**Abbreviations:** aHR, adjusted hazard ratio; CKD, chronic kidney disease; SGA, small for gestational age

$p < 0.001$) and diabetic CKD (aHR 1.96, 95% CI 1.56–2.47, $p < 0.001$), but did not persist for glomerular/proteinuric CKD.

When women who developed postpartum hypertension were excluded from analyses, the association with CKD was attenuated considerably (aHR 1.26, 95% CI 1.15–1.38, $p = 0.006$) (S11 Table).

## Discussion

This study aimed to determine whether women who experience HDP are at risk of CKD and whether this risk differs by CKD aetiology. Overall, preeclampsia was associated with significantly increased risk of CKD, and the time to CKD diagnosis was 2.7 years shorter in women who previously had preeclampsia than in those who did not. Women diagnosed with gestational hypertension were also at increased risk of CKD, but the strength of this association was less marked.

Women exposed to HDP had strongly increased risk of hypertensive CKD and diabetic CKD. These associations are consistent with previous cohort studies, which reported that preeclampsia is associated with increased risk of postpartum hypertension and type 2 diabetes [27–30]. Although these CKD subtypes were less commonly diagnosed in our sample than other forms of renal disease, they are likely to become predominant causes of CKD in an older cohort with longer follow-up. By contrast, we observed less marked associations between preeclampsia and tubulointerstitial CKD or nonspecific CKD.

Glomerular/proteinuric CKD accounted for one-third of all CKD cases. Preeclampsia was associated with a doubling in risk of glomerular/proteinuric CKD, and the median time to diagnosis was 3.5 years shorter. Preeclampsia may lead to glomerular endotheliosis, which results in glomerular dysfunction, podocyte loss [31, 32] and subsequent microalbuminuria [33, 34]. Notably, no significant association was observed for gestational hypertension and glomerular/proteinuric CKD. This lends support to the hypothesis that the association between preeclampsia and CKD may be mediated through persistent glomerular damage, possibly related to down-regulation of NRF-2 [12, 35], and not entirely through the effects of hypertension or hyperglycaemia.

We examined whether concurrent SGA impacted on associations between preeclampsia and CKD. Previous longitudinal studies have reported an increased risk of CVD [36–38] and ESKD [7] in women who had concurrent preeclampsia and SGA, but they used relatively broad composite outcomes. In our study, co-occurring SGA appeared to add to the risk of hypertensive CKD specifically, but it made little difference to the risk of other CKD aetiologies. Women who have both SGA and preeclampsia may experience more extreme placental dysfunction [39], and it is plausible that this signals a higher risk of hypertensive disease in later life [38]. Women who experienced SGA alone (without preeclampsia) were also at elevated risk of CKD, consistent with previous research [7, 40, 41], but the modest increases observed in our study suggest that this may be of limited clinical importance.

Preeclampsia was associated with higher risk of CKD in obese women compared with women whose pre-pregnancy BMI was normal. Previous studies of preeclampsia and maternal renal disease have either adjusted for obesity without considering the possibility of effect modification [15, 42] or have lacked any information on maternal BMI [7, 16, 41]. It is possible that women who develop preeclampsia have different cardio-renal risk profiles depending on their pre-pregnancy BMI. Pre-pregnancy obesity has been reported to be an independent risk factor for subsequent hypertension among women who ever experienced preeclampsia [43]. Furthermore, women may have elevated markers of long-term endothelial dysfunction if they were overweight before developing preeclampsia [44]. Although we restricted our analysis to women with complete information on antenatal BMI, we cannot rule out the possibility of unmeasured confounding from dyslipidaemia or recurrent preeclampsia, particularly if women with a first episode of preeclampsia received anti-hypertensive treatment or alternative cardio-protective intervention post partum.

Our findings support the need to optimise long-term follow-up of women exposed to HDP, and particularly high-risk women who experience preterm preeclampsia or recurrent preeclampsia. We did not have information on long-term blood pressure values for women in this study. It is uncertain whether screening for hypertension would suffice in preventing CKD or enabling earlier diagnosis of CKD in women with a history of preeclampsia. The additional value of screening for albuminuria is unknown and may depend on the underlying aetiology of CKD. However, it has been estimated that the number of patients with preeclampsia who need follow-up to detect one adverse event is about 4 for overt albuminuria and 157 for CKD, and the latter is likely to be a conservative overestimate [45]. Early renal damage may be masked by compensatory glomerular hyperfiltration [46], and this may also limit the ability to detect high-risk women until later in life. Thus, the optimal timing of engaging women in systematic renal and cardiovascular monitoring warrants further research. Nonetheless, 7% to 12% of all women will develop CKD in their lifetime [47–49], and the absolute risk of clinically significant disease is substantial.

## Strengths and limitations

To our knowledge, this is the largest study to investigate associations between preeclampsia and CKD to date and the first to report associations between gestational hypertension and

CKD subtypes. Its strengths include the use of national registry data with near-complete coverage and over 4 decades of follow-up [20]; classification of CKD according to specific aetiologies; adjustment for a broader range of covariates than previous studies, including maternal smoking and BMI; exclusion of women with a large number of relevant pre-existing comorbidities, as well as congenital and genetic forms of CKD to reduce confounding; and the use of time-dependent covariates.

However, the study is not without limitations. Although the NPR achieved national coverage for inpatients in 1987, outpatient data were only available from 2001 onwards, and the overall incidence of CKD was lower than expected. The SRR collected data on CKD from 2007 and is unlikely to be complete [50]. It is possible that cases of CKD were under-diagnosed or under-ascertained in the national registers; some women may have been too young to have developed symptomatic CKD despite their long follow-up time (median 21 years), and we cannot exclude the possibility of immortal time bias in our analysis. The NPR has high PPV for most diagnoses, but its sensitivity levels tend to be lower [51], and to our knowledge these parameters have not been formally measured for CKD or its subtypes. Thus, while those who were diagnosed with CKD in our dataset are likely to have valid diagnoses, the number of undiagnosed cases is uncertain.

Hypertensive CKD and diabetic CKD were less commonly diagnosed than was anticipated, and this may have reflected lower sensitivity levels for these diagnoses or relatively short median follow-up times, or it may reflect relatively low levels of obesity and dysglycaemia among Swedish women [52, 53]. The respective PPVs for ICD-8 coded diagnoses of pre-eclampsia and gestational hypertension were lower than for ICD-9 diagnoses [22], thus our overall results may be somewhat conservative. Our sensitivity analyses based on first deliveries after 1987 showed stronger associations between preeclampsia and CKD, but these were based on considerably fewer CKD cases, particularly when divided by CKD subtype.

Data on maternal BMI and smoking were incomplete and were only collected from 1982 onwards. We stratified by year of delivery in all our models and created a missing indicator variable to control for this. We cannot exclude the possibility of residual bias from using missing indicator variables. However, the results of our sensitivity analyses—when data on BMI were complete—were not substantially different.

Previous studies have controlled for postpartum hypertension when investigating associations between preeclampsia and cardiometabolic disease [16, 54], and despite an inherent risk of over-adjustment bias [55], the associations appear to persist. In our sensitivity analyses, associations with CKD persisted after excluding women with postpartum hypertension. This suggests that other non-hypertensive factors also contribute to the development of maternal CKD. This analysis was limited to those diagnosed with hypertension in hospital settings during follow-up and may have missed a large number of women who were diagnosed with hypertension in community settings. Further research is required to delineate the role of mediating factors, such as postpartum hypertension, hyperglycaemia, hyperlipidaemia, and changes in maternal BMI, in the association between HDP and maternal CKD.

## Conclusion

Preeclampsia is associated with an increased risk of maternal CKD in the years following pregnancy. This risk is higher after preterm preeclampsia, recurrent preeclampsia, or in preeclampsia complicated by pre-pregnancy obesity. The risk differs by CKD aetiology and is most marked for hypertensive CKD, diabetic CKD, and glomerular/proteinuric CKD. Gestational hypertension is also associated with elevated risk of CKD, although associations are more

modest than for preeclampsia. Women who experience HDP may benefit from systematic renal monitoring to prevent future CKD.

## Supporting information

**S1 Fig. Flow chart illustrating construction of study cohort.**
(DOCX)

**S1 Table. ICD codes used for disease definitions.**
(DOCX)

**S2 Fig. Kaplan-Meier survival curves for risk of CKD (overall and subtypes) among women whose first live birth occurred between 1973 and 2012 in Sweden, by exposure to preeclampsia.**
(DOCX)

**S2 Table. STROBE statement—checklist of items that should be included in reports of observational studies.**
(DOCX)

**S3 Table. Time to diagnosis of CKD subtypes among women whose first live birth occurred between 1973 and 2012 in Sweden, stratified by exposure to preeclampsia (*n* = 1,924,409).**
(DOCX)

**S4 Table. HRs for maternal CKD by history of preeclampsia and SGA, among women whose first live birth occurred between 1987 and 2012 in Sweden (*n* = 1,127,798).** HRs represent results of Cox regression models for associations between preeclampsia ± SGA and maternal CKD. Preeclampsia was a time-dependent variable. Fully adjusted models controlled for maternal age, country of origin, education level, parity, maternal BMI, smoking in pregnancy, exposure to gestational diabetes, and exposure to gestational hypertension. Models were stratified by year of delivery. *All $p < 0.001$. CKD, chronic kidney disease; HR, hazard ratio; SGA, small for gestational age.
(DOCX)

**S5 Table. HRs for maternal CKD by history of preeclampsia and preterm delivery, among women whose first live birth occurred between 1987 and 2012 in Sweden (*n* = 1,127,798).** HRs represent separate Cox regression models for associations between preeclampsia and maternal CKD. Preeclampsia was a time-dependent variable. Fully adjusted models controlled for maternal age, country of origin, education level, parity, maternal BMI, smoking in pregnancy, exposure to gestational diabetes, and exposure to gestational hypertension. Models were stratified by year of delivery. *All $p < 0.001$. **Exact number not reported as cell count $\leq$ 5. HR, hazard ratio; ne, not estimable; SGA, small for gestational age.
(DOCX)

**S6 Table. HRs for maternal CKD by history of recurrent preeclampsia, among women whose first live birth occurred between 1987 and 2012 in Sweden (*n* = 548,621).** *All $p < 0.001$. **Exact number not reported as cell count $\leq$ 5.
(DOCX)

**S7 Table. HRs for maternal CKD by history of preeclampsia and SGA, among women who first live birth occurred between 1973 and 2012 in Sweden with and without those who later developed postpartum hypertension (*n* = 1,924,409).** HRs represent separate Cox regression models for associations between preeclampsia and maternal CKD. Preeclampsia was a time-dependent variable. Fully adjusted models controlled for maternal age, country of

origin, education level, parity, maternal BMI, smoking in pregnancy, exposure to gestational diabetes, and exposure to gestational hypertension. Models were stratified by year of delivery. *All $p < 0.001$. HR, hazard ratio; SGA, small for gestational age.
(DOCX)

**S8 Table. HRs for maternal CKD by history of preeclampsia and preterm delivery, among women who first live birth occurred between 1973 and 2012 in Sweden with and without those who later developed postpartum hypertension ($n$ = 1,924,409).** HRs represent separate Cox regression models for associations between preeclampsia and maternal CKD. Preeclampsia was a time-dependent variable. Fully adjusted models controlled for maternal age, country of origin, education level, parity, maternal BMI, smoking in pregnancy, exposure to gestational diabetes, and exposure to gestational hypertension. Models were stratified by year of delivery. *All $p < 0.001$. **Exact number not reported as cell count $\leq$ 5. HR, hazard ratio; ne, not estimable; SGA, small for gestational age.
(DOCX)

**S9 Table. HRs for maternal CKD by history of recurrent preeclampsia, among women whose first live birth occurred between 1973 and 2012 in Sweden with and without those who later developed postpartum hypertension ($n$ = 1,924,409).**
(DOCX)

**S10 Table. HRs for maternal CKD by history of gestational hypertension, among women whose first live birth occurred between 1973 and 2012 in Sweden ($n$ = 1,924,409).** HRs represent separate Cox regression models for associations between preeclampsia and maternal CKD. Gestational hypertension was a time-dependent variable. Fully adjusted models controlled for maternal age, country of origin, education level, parity, maternal BMI, smoking in pregnancy, exposure to gestational diabetes, and exposure to preeclampsia. Models were stratified by year of delivery. HR, hazard ratio; SGA, small for gestational age.
(DOCX)

**S11 Table. HRs for maternal CKD by history of gestational hypertension, among women whose first live birth occurred between 1973 and 2012 in Sweden with and without those who later developed postpartum hypertension ($n$ = 1,924,409).** HRs represent separate Cox regression models for associations between preeclampsia and maternal CKD. Gestational hypertension was a time-dependent variable. Fully adjusted models controlled for maternal age, country of origin, education level, parity, maternal BMI, smoking in pregnancy, exposure to gestational diabetes, and exposure to preeclampsia. Models were stratified by year of delivery. HR, hazard ratio; SGA, small for gestational age.
(DOCX)

## Author Contributions

**Conceptualization:** Peter M. Barrett, Fergus P. McCarthy, Ali S. Khashan.

**Data curation:** Marius Kublickas, Ali S. Khashan, Karolina Kublickiene.

**Formal analysis:** Peter M. Barrett, Ali S. Khashan.

**Funding acquisition:** Peter M. Barrett, Karolina Kublickiene.

**Investigation:** Peter M. Barrett.

**Methodology:** Peter M. Barrett, Fergus P. McCarthy, Marie Evans, Peter Stenvinkel.

**Supervision:** Fergus P. McCarthy, Ivan J. Perry, Ali S. Khashan, Karolina Kublickiene.

**Validation:** Marie Evans, Marius Kublickas, Peter Stenvinkel.

**Writing – original draft:** Peter M. Barrett, Ali S. Khashan.

**Writing – review & editing:** Peter M. Barrett, Fergus P. McCarthy, Marie Evans, Marius Kublickas, Ivan J. Perry, Peter Stenvinkel, Ali S. Khashan, Karolina Kublickiene.

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
