## [Editor Report · Decision Letter 0]

6 Mar 2020

Dear Dr Barrett, 

Thank you for submitting your manuscript entitled "Hypertensive disorders of pregnancy and the risk of chronic kidney disease: a national registry-based cohort study" for consideration by PLOS Medicine.

Your manuscript has now been evaluated by the PLOS Medicine editorial staff and I am writing to let you know that we would like to send your submission out for external peer review.

Please re-submit your manuscript within two working days.

Kind regards,

Louise Gaynor-Brook, MBBS PhD

PLOS Medicine

---

## [Decision Letter · Decision Letter 1]

28 Apr 2020

Dear Dr. Barrett,

Thank you very much for submitting your manuscript "Hypertensive disorders of pregnancy and the risk of chronic kidney disease: a national registry-based cohort study" (PMEDICINE-D-20-00714R1) for consideration at PLOS Medicine. 

[LINK]

In light of these reviews, I am afraid that we will not be able to accept the manuscript for publication in the journal in its current form, but we would like to consider a revised version that addresses the reviewers' and editors' comments. Obviously we cannot make any decision about publication until we have seen the revised manuscript and your response, and we plan to seek re-review by one or more of the reviewers. 

We expect to receive your revised manuscript by May 19 2020 11:59PM. Please email us (plosmedicine@plos.org) if you have any questions or concerns.

We look forward to receiving your revised manuscript. 

Sincerely,

Emma Veitch, PhD

PLOS Medicine

On behalf of Clare Stone, PhD, Acting Chief Editor,

PLOS Medicine

plosmedicine.org

*The abstract should have a minor restructure, with the headings Background, Methods and Findings, Conclusions (Methods and Findings are a single combined section). 

*In the last sentence of the Abstract Methods and Findings section, would suggest including a brief description of the main limitation(s) of the study's methodology.

*At this stage, we ask that you include a short, non-technical Author Summary of your research to make findings accessible to a wide audience that includes both scientists and non-scientists. The Author Summary should immediately follow the Abstract in your revised manuscript. This text is subject to editorial change and should be distinct from the scientific abstract. Please see our author guidelines for more information: https://journals.plos.org/plosmedicine/s/revising-your-manuscript#loc-author-summary

*Currently, the STROBE guideline is used to support study reporting (and provided as supporting information), this is good, but would also suggest including a mention of this somewhere in the Methods section (eg stating - "This study is reported as per the Strengthening the Reporting of Observational Studies in Epidemiology (STROBE) guideline (then call out the supporting information file)". 

*Would also suggest clarifying in the Methods section of the paper whether the study had a prospective protocol or analysis plan. Please state this (either way) early in the Methods section.

Comments from the reviewers:

Reviewer #1: Review of PMED-D-20-00714-R1

This manuscript describes a study assess the effect of hypertensive disorders of pregnancy on incidence of chronic kidney disease. The long term effects of hypertensive disorders of pregnancy are important, and this manuscript offers a step forward in assessing these long term effects. However, I have several methodological concerns.

1. The study was limited to women having live births. When estimating the effect of HDP on CKD, this offers 2 challenges: 1) there may be shared predictors of live birth (vs still birth or miscarriage) and CKD, which mean that the incidence of CKD is higher/lower among women who have at least one live birth than the total population of women who have ever been pregnancy (which may alter the estimated HR); and 2) if HPD affect the probability of having a live birth AND there is a common cause of live birth and CKD, restricting the study population to those with live births will cause a collider stratification bias (see Cole SR, Platt RW, Schisterman EF, Chu H, Westreich D, Richardson D, Poole C. Illustrating bias due to conditioning on a collider. International journal of epidemiology. 2010 Apr 1;39(2):417-20.). To avoid the first concern, the manuscript should clearly state that the relevant target population of interest is women who have had at least one live birth, rather than all women who have been pregnant. To avoid the second concern, I recommend against using causal language.

2. The study excluded CKD diagnoses that occurred before a woman's last delivery. On page 7, line 178, the manuscript states that "we only considered women who were diagnosed with CKD at least 3 months after the last pregnancy." This raises several questions and concerns. First, what happened to women who had a CKD diagnosis and then had another pregnancy/delivery? Moreover, excluding these diagnoses essentially makes the time between first and last delivery "immortal", which opens the door to a variety of immortal time biases. For example, this approach may omit women experiencing such a strong effect of HDP that they are diagnosed with CKD just 6 months after their first delivery (if they later go on to deliver 2 more children later in life). Typically, any outcome definition that requires looking into the future to classify patients as cases or not opens the door to bias. This is particularly true if there may be common causes of future pregnancies and CKD.

3. Handling of competing events. Throughout, the analysis censored patients at death (I believe). If this is true, it should be stated explicitly. Moreover, a statement should be added to the definition of the parameter of interest and the interpretation of the results that the reported HRs are the HRs that would be seen could mortality be eliminated during the study period (see Austin PC, Lee DS, Fine JP. Introduction to the analysis of survival data in the presence of competing risks. Circulation. 2016 Feb 9;133(6):601-9). I also recommend presenting the results alongside results reporting overall mortality during the study period and, if overall mortality is not negligible, mortality by exposure group.

Specific points:

4. Page 4, line 113: here, or in the statistical analysis section, please make it clear that deaths were censored (or clarify if this was not the case).

5. Page 4, line 115: here, or in the results, please report how many were excluded based on each criterion.

6. Page 6, line 160: minor point, but the parenthetical here I believe should read (32 weeks to 36 weeks + 6 days) to avoid confusion

7. Page 6, line 165: please add a sentence of phrase here stating that this analysis was conducted only among women with exactly 2 deliveries. I was left wondering what the exposure definition was for women with >2 deliveries and did not understand that this was conducted among a subset until the Results.

8. Page 7, line 189: how were covariates selected?

9. Page 7, line 195: the missing indicator method to account for missing data has been shown to produce biased results in many settings (see e.g., Greenland S, Finkle WD. A critical look at methods for handling missing covariates in epidemiologic regression analyses. American journal of epidemiology. 1995 Dec 15;142(12):1255-64.). Are there features of the current analysis that lead us to believe that it will work well here? If so, those should be stated.

10. Page 8, line 200: from my understanding of the exposure definitions, women were considered to have gestational hypertension if they had a subset of the criteria for preeclampsia. If this is true, wouldn't these variables be perfectly collinear?

11. Page 8, line 210: in models 2 through 4, the model form should be explained. Specifically, how was SGA modeled in model 2? Did the models include interaction terms between preeclampsia and SGA / preterm delivery?

12. Page 9, line 239: how many CKDs were excluded because they occurred before woman's last delivery?

13. Table 1: Is it correct to assume those in the "Preeclampsia" column were those who ever had a diagnosis of preeclampsia? If so, this should be noted (and if not, it should be clarified).

14. Page 12, line 251: should the first sentence read "Women who had ever experienced preeclampsia…" (rather than "ever experienced") to make it clear that preeclampsia exposure was time varying?

15. Page 21, line 357: should this be "incidence" rather than "prevalence"?

Reviewer #2: This is a well written paper in a Swedisch cohort where by this group the many other analysis have been done on ESRD, CKD in relation to prematurity. I do think that the paper is interesting but question whether is it novel enough in addition to the other papers. 

1. They do not have long-term blood pressure value of patients and therefor can not conclude that screening for renal function or albuminuria would have any additional value after preeclampsia. Screening for hypertension after preeclampsia and correcting for this while making statement on renal follow-up is essential

2. Mechanisms of renal damage after preeclampsia can be multi-factorial. So far only marginal difference in sFlt1 are found as reviewed by Hypertension. 2016;68:00-00. DOI: 10.1161/HYPERTENSIONAHA.116.07907. NRF-2 is not a well know factor associated with glomerular damage in preeclampsia. Changes in the RAAS system, metabolic system and other factors causing endothelial dysfunction could be involved. 

Reviewer #3: This is an interesting and comprehensive analysis which has examined the long-term risk of CKD in women who developed hypertensive disorders in pregnancy. It represents a large and well curated dataset and provides important information which will inform the care and counselling of women who develop pregnancy complications. I have a number of questions listed below which I think could be clarified in the manuscript. Overall the data is well-described and clearly presented.

Why were still births excluded? I accept that sometimes difficult to know the cause but stillbirths in the context of preeclampsia/SGA (approx. 40% of stillbirths) should be included?

What was the comparator group(s) in the recurrent PE model? Are women with more than two pregnancies excluded from this analysis - needs clarification.

Definition (coding) of gestational hypertension? - women who only developed hypertension immediately after birth (ie within 6 weeks) or women who developed hypertension anytime after or between pregnancies? It would seem from the discussion that there is a distinct possibility that the diagnosis of post partum hypertension is under reported. This under reporting may significantly attenuate the association between pregnancy disease and CKD specifically - ie the association may be attributable to persistent hypertension (and/or diabetes) than pre-eclampsia per se? Should this perhaps be acknowledged in the abstract?

Whilst I understand the reason for excluding baseline hypertension diagnoses from the analysis - I wonder whether if this data is available whether it would be useful to compare the long term risk of CKD in this group of women which would provide a useful context comparison to the rates of CKD in women with de novo hypertension developing in pregnancy. I suspect this data is not available though?

Why do such a large number of CKD cases have non-specified CKD. Does the coding account for AKI related to another event which subsequently resolves?

Are the differences in time course significant? Should the different patterns of presentation of subsequent CKD be considered in follow up programmes?

Line 343 in discussion - what follow up would be required. Why do the authors consider this to be a conservative over estimate?

Line 344 - 7-12% in their lifetime population risk presumably includes a large amount of CKD in older age >30years away from pregnancies and not covered by the time frame of the current analysis. The lifetime risk discussed here should be the population risk within the relevant time periods covered by the analysis presented which is not 7-12%?

Consider changing "non-pre-eclamptic" to women who had never had pre-eclampsia? Same for "pre-eclamptic" as a term in general.

[LINK]

---

## [Decision Letter · Decision Letter 2]

26 Jun 2020

Dear Dr. Barrett,

Thank you very much for re-submitting your manuscript "Hypertensive disorders of pregnancy and the risk of chronic kidney disease: a national registry-based cohort study" (PMEDICINE-D-20-00714R2) for review by PLOS Medicine.

I have discussed the paper with my colleagues and the academic editor and it was also seen again by reviewers. I am pleased to say that provided the remaining editorial and production issues are dealt with we are planning to accept the paper for publication in the journal.

[LINK]

We look forward to receiving the revised manuscript by Jul 03 2020 11:59PM. 

Sincerely,

Clare Stone, PhD

Managing Editor 

PLOS Medicine

plosmedicine.org

Requests from Editors:

Title – please add the country setting

Abstract and throughout, please provide p values with quantifiable data and where 95% CIs are given

Please be more explicit in the abstract about the limitations of your study starting with ‘Limitations of the study are….’

Please use square brackets for refs in the main text.

Please break the supp files into separate ones and the STROBE needs to be provided with sections and paragraphs instead of page numbers

At line 74, "In this study, we found that ... were associated ..." or similar. 

At line 102, "Our findings suggest that ..." or similar. 

Methods, was there a protocol or prespecified analysis plan?

Comments from Reviewers:

Reviewer #1: Overall, this revision is very responsive to my previous comments. A few minor points are added below:

1. Page 11, line 286: Presumably this is time to CKD diagnosis *after first live birth*?

2. Page 6, line 174: I found this sentence confusing. I thought the current study included all women who had live singleton births during/after 1973, not limited to those with preeclampsia after 1973. I recommend rewording this sentence to clarify that analyses were repeated after restricting the entire study population to those with a first live birth from 1987 onwards.

3. Page 9, line 234: I recommend clarifying that "missing indicators were used to account for missing data on smoking and BMI" (as opposed to ending the sentence with "this"). As a side note, I accept the authors' logic that, because results were similar after sensitivity analysis that limited the study population to 1987 and after, when there was less missing data, the missing indicator method was unlikely to have induced substantial bias. However, I still have concerns about use of the missing indicator method in this setting and recommend adding language to the discussion acknowledging the potential for (or at least possibility of) residual bias when using this approach.

Reviewer #3: Comments considered and addressed.

The manuscript has been significantly improved, I have no further suggestions/comments.

[LINK]

---

## [Editor Report · Decision Letter 3]

15 Jul 2020

Dear Dr Barrett, 

On behalf of my colleagues and the academic editor, Dr. Jenny Myers, I am delighted to inform you that your manuscript entitled "Hypertensive disorders of pregnancy and the risk of chronic kidney disease: a Swedish registry-based cohort study" (PMEDICINE-D-20-00714R3) has been accepted for publication in PLOS Medicine. 

PRODUCTION PROCESS

PRESS

PROFILE INFORMATION

Thank you again for submitting the manuscript to PLOS Medicine. We look forward to publishing it. 

Best wishes, 

Clare Stone, PhD

Managing Editor 

PLOS Medicine

plosmedicine.org